# Geographical Erasure in Language Generation

**Pola Schwöbel**
Amazon
Berlin, Germany
schwobel@amazon.de

**Jacek Golebiowski**
Amazon
Berlin, Germany
jacekgo@amazon.de

**Michele Donini**
Amazon
Berlin, Germany
donini@amazon.de

**Cédric Archambeau**[*]
Helsing
Berlin, Germany
cedric.archambeau@helsing.ai

**Danish Pruthi**[*]
Indian Institute of Science (IISc)
Bangalore, India
danishp@iisc.ac.in

## Abstract

Large language models (LLMs) encode vast amounts of world knowledge. However, since these models are trained on large swaths of internet data, they are at risk of inordinately capturing information about dominant groups. This imbalance can propagate into generated language. In this work, we study and operationalise a form of *geographical erasure*, wherein language models underpredict certain countries. We demonstrate consistent instances of erasure across a range of LLMs. We discover that erasure strongly correlates with low frequencies of country mentions in the training corpus. Lastly, we mitigate erasure by finetuning using a custom objective.[1]

## 1 Introduction

Large pretrained models serve as base models for many downstream NLP applications, including question-answering, dialogue, common-sense reasoning, classification, tagging, translation, summarisation, and generation (Devlin et al., 2018; Brown et al., 2020; Chowdhery et al., 2022). Despite their increasing utility, there are concerns about how they reflect and amplify biases in the training data. For instance, unfiltered data originating from the internet is known to be rife with toxic, misogynistic, and stereotyping content. Many studies highlight biases in model outputs, primarily concerning *representational harms* (Barocas et al., 2017), where a section of society (e.g., women, LGBTQ+ communities) are represented in poor light, or are ignored by the system (Bolukbasi et al., 2016; Guo and Caliskan, 2021; May et al., 2019;

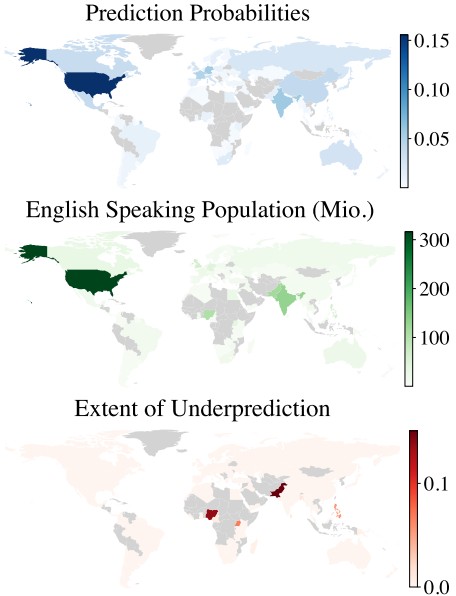

Figure 1: **Some countries are vastly underpredicted compared to their English speaking populations.** *Top:* Country probabilities assigned by GPT-NeoX when prompted with "I live in". *Middle:* English speaking populations per country. *Bottom:* Countries experiencing erasure, i.e. underprediction compared to their population by at least a factor 3 (see §3). Data is missing for grey countries (see §6).

Tan and Celis, 2019). While important, such studies predominantly examine biases related to race, gender, occupation and sexual orientation.

An important—and often overlooked—aspect of inclusive model development is *geographical* inclusion. This is particularly important at a time when most large-scale model training efforts come from a small set of regions. Further, these models are trained using internet data, whose access, in the first place, is unequally distributed (Blank, 2017; Center, 2021). Minimising cultural and geo-

---

[*]Work done while at Amazon.

[1]Code available at https://github.com/amazon-science/geographical-erasure-in-language-generation.

graphic identities is referred to as erasure (Roche, 2019) and is studied by linguists and social scientists in the context of imperialism and colonialism, where "people are silenced in the historical record [...], their contemporary presence rendered invisible, and their existence written out of the future" (Roche, 2019). Automated systems and their developers exclude certain groups unintentionally, but the risk of being "written out of the future" remains pressing: the produced content is fed back into the internet. In lifelong learning setups, the generated content becomes the training data of tomorrow's models, closing the vicious circle of reinforcing social hierarchies (see §4.5 of (Sheng et al., 2021)).

In this paper, we reveal instances of *geographical erasure*, wherein models underpredict geographical regions (see Fig. 1). For instance, GPT2 assigns nine times higher likelihood to "I live in Canada" than "I live in Pakistan", whereas Pakistan's English-speaking population is almost four times to that of Canada. By comparing model outputs with population statistics, we operationalise geographical erasure (§3). Using this measure, we first demonstrate the existence of erasure for several countries across different prompt formulations (§4.1). Studying the consistency across a range of language models from the GPT and LLaMA model families, we find that several countries — Nigeria, Pakistan, Eswatini, Uganda and Madagascar — are affected by erasure under *all* models (§4.2). Following related work (Lin et al., 2021; Rae et al., 2021; Nadeem et al., 2020), we study the impact of model size on the extent of erasure, and find it to be small — erasure occurs across all sizes (§4.3). To identify the causes of erasure, we compute the unigram frequencies of countries in the training corpus (§4.4). They closely match our model predictions indicating that the composition of training data is a main source of erasure. Lastly, we alleviate erasure via supervised finetuning. We study the impact of mitigation on generation quality as measured in perplexity on Wikitext-2-v1. Our finetuning strategy proves to be an effective mitigation mechanism which generalises and has small impact on generation quality (§4.5).

## 2   Related Work

The literature on fairness in machine learning distinguishes between representational and allocational harms (Barocas et al., 2017; Crawford, 2017; Blodgett et al., 2020). Allocational harms concern the unfair distribution of resources, e.g. when a group is denied bank loans disproportionally by an automated system. Allocational harms tend to be more easily measured through standard fairness metrics like demographic parity (Dwork et al., 2012) and equality of opportunity (Hardt et al., 2016). Those do not directly apply to open-ended generation tasks, where we instead study representational harms, which arise when a system "represents some social groups in a less favourable light than others, demeans them, or fails to recognise their existence altogether" (Blodgett et al., 2020); the last case being the focus of our work.

*Fairness measures for language generation* usually define bias as differences between demographic groups (Sheng et al., 2021). For example, Dhamala et al. (2021) find that female pronouns are more likely to elicit positive text from an LLM than male pronouns. Similarly, Huang et al. (2019) compare different occupations, names and countries on produced sentiments. Nangia et al. (2020) and Nadeem et al. (2020) compare the probability of stereotypical and non-stereotypical sentences under a model in order to measure whether it encodes stereotypes against different demographic groups. Along the same lines, the WinoGender test (Rudinger et al., 2018) measures gender biases in co-reference resolution tasks. Taking a distributional view similar to our work, Rae et al. (2021) investigate biases in the context of occupation, however, they again compare predictions for different genders with each other. In general, such comparative bias tests are well-adapted by authors proposing new models (Touvron et al., 2023; Rae et al., 2021; Hoffmann et al., 2022; Scao et al., 2022). Instead of comparing model predictions against each other, we compare model predictions to *real world ground truth distributions* in order to quantify bias.

*Ground truth-based measures* are not commonly used as a metric for fairness but important when evaluating a model's truthfulness. Petroni et al. (2019) and Lin et al. (2021) provide datasets of real world facts against which to benchmark LLMs' knowledge. Similar to our work, Zhou et al. (2022) measure the frequency of country predictions, and how underprediction correlates with a country's GDP. Contrary to their count-based approach we propose a more fine-grained metric for erasure and extend the analysis to auto-regressive models. Unlike theirs, our erasure metric can be employed as a loss function for finetuning, to specifically miti-

gates erasure. Liang et al. (2022) propose a similar metric for erasure in the domains of gender and race. Like us, they compare model to ground truth distributions, though they measure a total variation distance where we use a KL-divergence based approach (see §3.3). The authors assume uniform ground truth whereas we construct a domain specific distribution (see §3.2). Lastly, unlike ours, their analysis does not cover any mitigation efforts. Similar in spirit, geographical representativeness has been studied for text-to-image generation models (Rojas et al., 2022; Basu et al., 2023).

# 3 Method

Our goal is to measure, and later mitigate, the extent to which large pretrained models underpredict some countries when generating language. We formalise this notion here. Note that while we are studying autoregressive models in this work, the methodology extends straightforwardly to masked models. Similarly, we are interested in measuring and reducing geographical erasure, but the analysis can be applied to other attributes where ground truth is available. For example, one could measure erasure with respect to age, ethnicity, religion or gender using the same formalism.

## 3.1 Obtaining Model Predictions

Let $p$ be our language model over vocabulary $\Omega$. We consider open-ended generation tasks for autoregressive models. Such models predict the next token given previous ones, i.e. for a sequence of $L$ tokens $x^{1:L} \subset \Omega$ the probabilities factorise as

$$p(x^{1:L}) = \prod_{k=1}^{L-1} p(x^{k+1}|x^{1:k}). \qquad (1)$$

We use pretrained models and condition on a short prompt, or context, of variable length $L$: $c = x^{1:L}$. Given this prompt, we compute the predictive distribution over a set of $M$ candidates $\{x_i\}_{i=1}^M = \mathcal{X} \subset \Omega$; see §3.2 for how these $M$ countries are chosen. For a candidate country $x_i \in \mathcal{X}$ we compute $p(x_i|c)$ as

$$p(x_i|c) = \frac{p(x_i, c)}{p(c)} = \frac{p(x_i, c)}{\sum_{x \in \mathcal{X}} p(x, c)}, \qquad (2)$$

i.e., we compute $p(\text{"I live in } x_i\text{"})$ for all candidate countries $x_i$ and normalise. If a country is tokenised into multiple tokens, $x_i = x_i^{1:J}$, we multiply the probability of the $J$ subtokens according to (1). As before, superscript indicates position and subscript indicates the country name,

e.g., $x_7 = $ "Uganda" is tokenised into $x_7^0 = $"U", $x_7^1 = $"g", $x_7^2 = $"anda". As a consequence, $p(x_i|c)$ tends to be smaller for multi-token country names. Concerningly, Zhou et al. (2022) show that this issue predominately impacts low GDP-countries.

Some countries in $\mathcal{X}$ are referred to by more than one name, e.g., "UK" and "United Kingdom". We disambiguate the countries using a list of alternative names[2] to obtain the final $p(x_i|c) = \sum_{a \in \mathcal{A}} p(x_i^a|c)$ for all alternative names $x_i^a$.

In the following sections, we sometimes write $p(x_i|c) = p_i$, omitting the dependency on the prompt unless ambiguous. Note that we work directly on the model probabilities and discuss the impact on generated language in §6.

## 3.2 Obtaining Ground Truth

To measure erasure, we compare the generation distribution (given by equation 2) to a ground truth distribution $p^{\text{true}}$ over the candidate countries, writing $p^{\text{true}}(x_i) = p_i^{\text{true}}$ as before. The ground truth is given by real world data, i.e., we compare our predictions to the actual population of country $x_i$. We adjust for the fact that our models are trained on English texts only by considering *English speaking* populations as ground truth (see §6 for limitations of this approach). The number of English speakers per country is obtained from a Wikipedia list containing data for $M = 127$ countries at the time of writing—we use these 127 countries for our analysis.[3] Unlike the model predictions $p(x_i|c)$, the ground truth $p^{\text{true}}(x_i)$ is prompt-independent. We will generalise model predictions to be prompt-independent as well by marginalising prompts in §3.5. See Figure 2 (left) for an example of model predictions and ground truth.

## 3.3 Measuring Erasure

With these prerequisites in place, we can now formalise erasure using the relationship of predictive distribution and ground truth.

**Definition 1** (Erasure Set). *For a ratio threshold $r > 1$ we define the erasure set under model $p$, ground truth $p^{true}$ and prompt $c$ as*

$$\mathcal{S}_r^c = \left\{ x_k : \frac{p^{\text{true}}(x_k)}{p(x_k|c)} > r \right\}. \qquad (3)$$

---

[2]List of alternative country names from https://en.wikipedia.org/wiki/List_of_alternative_country_names, retrieved on Sept. 26th, 2023.

[3]From https://en.wikipedia.org/wiki/List_of_countries_by_English-speaking_population, retrieved on Sept. 26th, 2023.

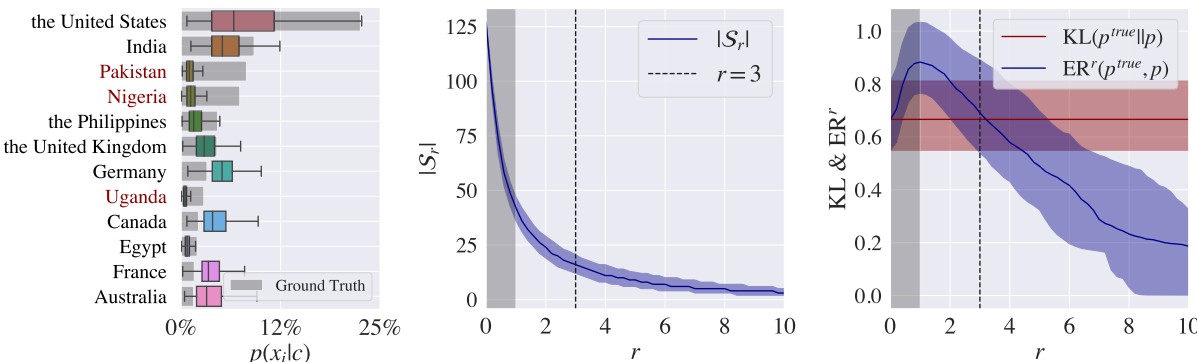

Figure 2: **Understanding erasure.** *Left:* OpenLLaMA, 7B vastly underpredicts the occurrence of Pakistan, Nigeria and Uganda. We plot country predictions given prompts $p(x_i|c)$ for different re-phrasings of the prompt "I live in" (boxplots) and ground truth (barplot, grey). Country names experiencing erasure ($x_i \in \mathcal{S}_3$, see §3.3) are in red. We show the 12 countries with the largest English speaking populations (in decreasing order). *Middle:* Erasure set size $|\mathcal{S}_r|$ as a function of $r$ for OpenLLaMA, 7B. We plot the median (solid line) and $25^{th} - 75^{th}$ percentiles (blue shaded area) over different rephrasings (see §3.5) of the same prompt. The dashed line marks $r = 3$, the threshold we use in the experiments. This choice is further motivated in §3.4. *Right:* Comparing $\mathrm{ER}^r$ (blue) for different $r$ to the KL-divergence (red). We pick $r = 3$, the integer value for which KL and $\mathrm{ER}^r$ are the most similar.

In the example in Figure 2 (left), we prompt the OpenLLaMA model with different versions of the prompt "I live in" and aggregate the predictions (see §3.5 for rephrasing and aggregation). We then compute the erasure set for $r = 3$, i.e. countries that are three times more prevalent in the ground truth than in our predictions. We obtain $\mathcal{S}_3 = \{$Pakistan, Nigeria, Uganda, Bangladesh, Iraq, Madagascar and Eswatini$\}$. A simple metric of erasure is the size of the erasure set $|\mathcal{S}_r|$ for a user-specified $r$, here, $|\mathcal{S}_3| = 7$.

$|\mathcal{S}_r|$ measures *how many* countries are "erased" (underrepresented by at least factor $r$). To obtain a more fine grained numerical evaluation we measure *by how much* they are underrepresented compared to ground truth by reporting the following metric.

**Definition 2** (Erasure). *Erasure w.r.t. ground truth $p^{true}$ at threshold $r$ is defined as*

$$\mathrm{ER}^r(p^{\text{true}}, p) = \sum_{i \in \mathcal{S}_r} p_i^{\text{true}} \log\left(\frac{p_i^{\text{true}}}{p_i}\right). \quad (4)$$

### 3.4 Properties of $\mathrm{ER}^r$

A careful conceptualisation of any proposed fairness metric is crucial (Schwöbel and Remmers, 2022; Blodgett et al., 2020). We motivate our definition of $\mathrm{ER}^r$ here. Firstly, if $p = p^{\text{true}}$ then $\mathrm{EB}^r(p^{\text{true}}, p) = 0$ for all $r$; i.e., no erasure occurs when the distributions match. Secondly, unlike the total variation distance suggested in Liang et al. (2022), we want our metric to be sensitive to *relative* rather than absolute errors, so that countries

with small populations are also taken into account. Hence we report (log-)ratios in the definition of $\mathrm{ER}^r$ (4). On the other hand, while we believe this sensitivity to less-populated countries is important, we do acknowledge that underpredicting big ground truth populations is particularly harmful as it impacts a lot of users. Thus, we weight the log-ratios by the ground truth probabilities $p_i^{\text{true}}$.

A third factor in our choice of metric is the close relation of (4) and the KL-divergence $\mathrm{KL}(p^{\text{true}}||p)$. $\mathrm{ER}^r$ is an additive component of the KL-divergence:

$$\mathrm{KL}(p^{\text{true}}||p) = \sum_{i \in \mathcal{S}_r} p_i^{\text{true}} \log\left(\frac{p_i^{\text{true}}}{p_i}\right) + \quad (5)$$
$$\sum_{i \in \mathcal{X} \setminus \mathcal{S}_r} p_i^{\text{true}} \log\left(\frac{p_i^{\text{true}}}{p_i}\right).$$

This close relation to a well-defined divergence measure allows for theoretical analysis and helps practitioners build on existing intuitions.

**The choice of r** is a crucial hyperparamter, as $|\mathcal{S}_r|$ and $\mathrm{ER}^r$ are defined in terms of $r$. We discuss the impact here and visualise it in Figure 2 (middle and right). For small values of $r$, we include all terms in (5), i.e.,

$$\lim_{r \to 0} \mathcal{S}_r = \mathcal{X} \text{ and } \lim_{r \to 0} \mathrm{ER}^r(p^{\text{true}}, p) = \mathrm{KL}(p^{\text{true}}||p).$$

For larger values of $r$, we instead have

$$\lim_{r \to \infty} \mathcal{S}_r = \emptyset \text{ and } \lim_{r \to \infty} \mathrm{ER}^r(p^{\text{true}}, p) = 0.$$

See Figure 2 (right) for this relationship. Since we want to measure erasure or underprediction, we study cases where $p^{\text{true}} > p$, i.e., for values $r > 1$.[4] We pick $r$ to be an integer such that $\text{ER}^r(p^{\text{true}}, p) \approx \text{KL}(p^{\text{true}} || p)$, that is $r = 3$ in the experiment in Fig. 2 (right). We find that this value is the same across all our models (see Appendix A), so we choose $r = 3$ globally. This choice of $r$ is based on a mathematical heuristic. An alternative way of choosing this parameter might be implied by legal or ethical constraints. For example, a guideline on adverse impact by the US Equal Employment Opportunity Commission (1979) defines "a substantially different rate of selection" at $80\%$. In this labour market use case, $r = 1/0.8 = 1.25$ would be the corresponding hyperparameter.

**Differentiability** is an important property of our metric since we want to use it for finetuning LLMs in §4.5. For fixed $r$, $\text{ER}^r$ is differentiable almost everywhere (with respect to the network weights). Singularities occur at those points that add new countries to the erasure set $\mathcal{S}_r$ in (3), i.e., weights such that $p_k^{\text{true}} = p_k$ for any country $k$.

### 3.5 Prompt Rephrasing

The erasure set definition in (3), and consequently the notion of erasure in (4) are prompt-dependent. However, we are interested more generally in the model's world knowledge rather than its completion of a specific prompt. Hence, we would like to aggregate the effect over all prompts encoding the meaning $\mathcal{M} =$"home country", by using the following marginal distribution:

$$p(x|\mathcal{M}) = \int p(x|c)p(c|\mathcal{M})\mathrm{d}c. \qquad (6)$$

The relationship between a prompt $c$ and its meaning $\mathcal{M}$ is complex, hence computing (6) is intractable. Here, we will rely on simple, pragmatic techniques to semi-automatically construct a set of sample prompts $\mathcal{D} \sim P(c|\mathcal{M})$ from a seed prompt $\tilde{c}$. We rephrase $\tilde{c}$ while preserving its meaning to generate additional prompts. This is common practice: Jiang et al. (2020) use mining- and translation-based paraphrasing methods while Romano et al. (2006) rely on templates for paraphrasing. In light

---

[4]There is a degree of symmetry in our measurement: being probability distributions, $p^{\text{true}}$ and $p$ sum to one. Thus, when $p^{\text{true}} > p$ for $\mathcal{S}_r$, there are other countries for which the opposite is true, i.e. that are overpredicted. In general, we believe underprediction to be more likely to cause harms than overprediction (see §1), hence we focus on measuring erasure.

of recent advances in LLMs, another way to automatically rephrase prompts is by using a model that has been finetuned for paraphrasing (Niu et al., 2020). Even simpler, we use an off-the-shelf model by prompting ChatGPT to rephrase the $\tilde{c} =$"I am from" seed prompt.[5] After manually removing irrelevant prompts we obtain 16 base formulations. We further expand the set of prompts by replacing sentence subjects. For example, we expand "I live in" into {"You live in", "He lives in", "She lives in", ...,}, producing a total of $|\mathcal{D}| = 955$ prompts. Details and a list of all prompts can be found in Appendix B. We use the dataset of 955 prompts to approximate the marginal in (6) assuming different prior probabilities $p(c|\mathcal{M})$ as follows:

**(1) Uniform prompt distribution:**

$$p(c|\mathcal{M}) = \frac{1}{|\mathcal{D}|}, \text{ then} \qquad (7)$$

$$p(x|\mathcal{M}) \approx \frac{1}{|\mathcal{D}|} \sum_{c \in \mathcal{D}} p(x|c) = p^{\text{agg\_uni}}(x|\mathcal{M}). \qquad (8)$$

**(2) Model-induced prompt distribution:**

$$p(c|\mathcal{M}) = \frac{p(c)}{\sum_{c \in \mathcal{D}} p(c)} \qquad (9)$$

where $p(c)$ is the probability given by the autoregressive language model (1). In this case,

$$p(x|\mathcal{M}) \approx \sum_{c \in \mathcal{M}} p(x|c)p(c|\mathcal{M})$$
$$= p^{\text{agg\_model}}(x|\mathcal{M}). \qquad (10)$$

## 4 Experiments

In this section, we show the existence of geographical erasure across different LLMs and different prompt wordings (§4.1). We highlight the consistency of erased countries across models (§4.2) and investigate the impact of model size on erasure (§4.3). We identify possible causes of erasure (§4.4) and explore a mitigation strategy (§4.5).

**The models under consideration** are GPT2 (Radford et al., 2019), 117M, 345M, 774M and 1.6B weight versions, GPT-Neo (Black et al., 2021), 125M, 1.3B and 2.7B weight versions, GPT-NeoX, 20B weights (Black et al., 2022) and open source reproductions of the LLaMA model (Touvron et al., 2023; Geng and Liu, 2023; Computer, 2023), 3B and 7B weights. We obtain all implementations from HuggingFace.[6]

---

[5]Accessed via `https://chat.openai.com/`.
[6]Via `https://huggingface.co/docs/transformers`.

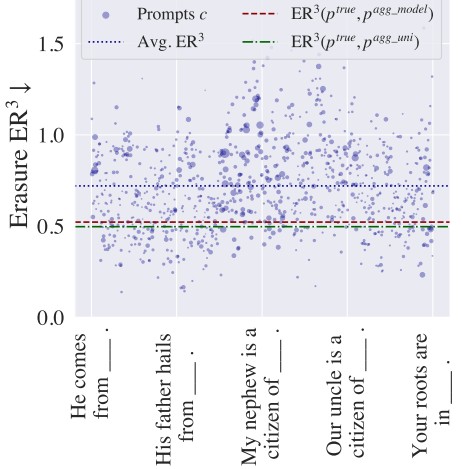 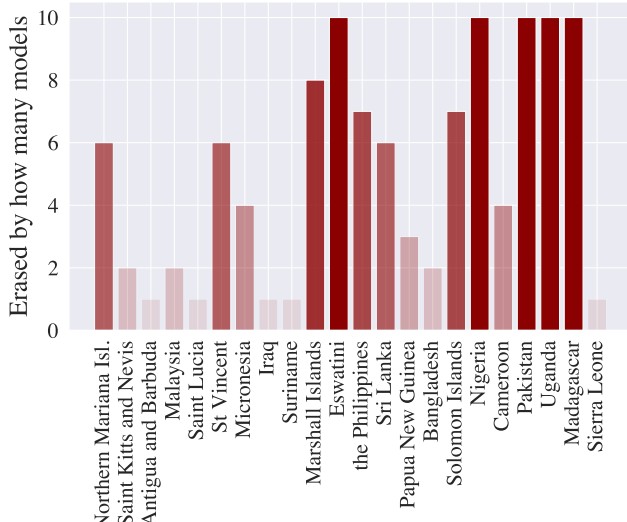

Figure 3: **Geographical erasure occurs for all prompt rephrasings, and many countries experience erasure consistently under all models.** *Left:* OpenLLaMA, 7B results for 955 individual prompts (blue dots) along the x-axis, with some example prompts as axis labels. We also plot $ER^r$ in aggregate: The blue line is the average over individual prompts $\frac{1}{C} \sum_c ER^3(p^{\text{true}}, p(\cdot|c))$, green is the uniform aggregate $ER^3(p^{\text{true}}, p^{\text{uni\_agg}})$ (8) and red is the model-induced aggregate $ER^3(p^{\text{true}}, p^{\text{model\_agg}})$ (10). Size of dots corresponds to the probability assigned to the respective prompt under the model. The gap between blue and red/green aggregations is explained in §3.5. *Right:* Of the $M = 127$ countries, 105 do not experience erasure at $r = 3$ for any of the models. For the remaining 22, we plot model counts here. Bars are coloured according to counts and sorted by GDP per capita (decreasing from left to right). We use aggregated predictions according to Equation 10.

## 4.1 Impact of Prompt Wording

We start by investigating how dependent erasure is on the exact phrasing of the prompt. We prompt the models with rephrased versions of "I live in" (see §3.5) and compute erasure $ER^3(p^{\text{true}}, p(\cdot|c))$ for each prompt $c$. In Figure 3 (left), we plot the (i) erasure for individual prompts (dots); (ii) the average erasure $\frac{1}{C} \sum_c ER^3(p^{\text{true}}, p(\cdot|c))$ denoted by the blue dotted line; (iii) erasure for the uniform marginal distribution from (8) using a green dashdotted line; and (iv) erasure for the model-induced marginal distribution from (10) as a red dashed line. The size of the blue dots indicates $p(c|\mathcal{M})$.

The magnitude of erasure $ER^3(p^{\text{true}}, p(\cdot|c))$ differs across the phrasings $c$, however, *erasure exists in all versions* (that is, $ER^3 > 0$ with p-value $\ll 0.01$). We note that erasure under the aggregate distribution is smaller than the average erasure ($ER^3(p^{\text{true}}, p^{\text{agg\_uni}}) < \frac{1}{C} \sum_c ER^3(p^{\text{true}}, p(\cdot|c)$ in Figure 3 (left)). This follows from Jensen's inequality (see Appendix C for details). Throughout the remainder of the paper, we will report the aggregates from (8) and (10) along with boxplots of $ER^3$ to account for the variance due to rephrasings.

## 4.2 Who is Experiencing Erasure?

We evaluate whether the same countries experience erasure under all the examined 10 models, and what characterises these countries. Out of the $M = 127$ countries under analysis, 105 do not experience erasure at $r = 3$ for any of the models. For the remaining 22 nations, Figure 3 (right) shows the number of models by which they are erased. Worryingly, Eswatini, Nigeria, Pakistan, Uganda and Madagascar experience erasure under *all* 10 analysed models. The x-axis in Figure 3 (right) is ordered by GDP per capita, in decreasing order from left to right.[7]

## 4.3 Impact of Model Size

Related work (Lin et al., 2021; Rae et al., 2021; Nadeem et al., 2020) reports mixed results on the relationship between model size and bias. On the one hand, Lin et al. (2021) report that on the TruthfulQA benchmark, "[l]arger models are less truthful". This is because large models surface the common human misconceptions that the questions are designed to elicit. Such misconceptions are likely present in the training data which the larger mod-

---

[7]Data from `https://en.wikipedia.org/wiki/List_of_countries_by_GDP_(nominal)` on Sept. 26th, 2023.

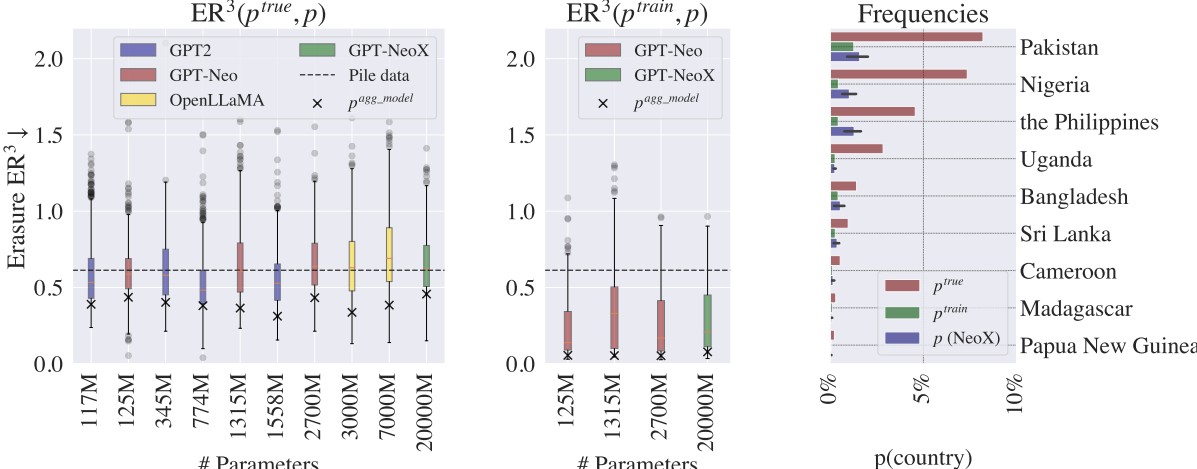

Figure 4: **Erasure in models closely matches distribution of country mentions in training data.** *Left:* Geographical erasure or GPT-type models of different sizes. Size is on the x-axis (axis not to scale). Blue models are GPT2, red models are GPT-Neo, yellow are OpenLLaMA and the green model is GPT-NeoX, crosses below each box plot are aggregated results over different prompts (see Eq. 10). The dashed line indicates $ER^r(p^{\text{true}}, p^{\text{train}})$. *Middle:* Geographical erasure for GPT-type models of different sizes, assuming training frequency as the ground truth (instead of world population). *Right:* Ground truth (red) and Pile training data (green) distributions compared to GPT-NeoX predictions (blue) on countries in the erasure set $\mathcal{S}_3$ (of GPT-NeoX w.r.t. ground truth).

els match more faithfully. Similarly, Nadeem et al. (2020) find that the larger models exhibit more stereotyping, again this is probably because they match stereotypes in the training data more closely. On the other hand, Rae et al. (2021) "do not see a consistent correlation between model size and bias" in their tests for gender-occupation bias.

We visualise the extent of geographical erasure with varying model sizes in Figure 4 (left). Like Rae et al. (2021), we do not find model size to have a big impact. We hypothesise that even the smaller models closely mimic the frequency distribution (of country mentions) in the training corpus, similar to Rae et al. (2021)'s experiment. We believe that this is not the case in the test by Lin et al. (2021) and Nadeem et al. (2020), because their tests go much beyond unigram frequencies, and smaller models do not exhibit such subtle biases. We explore the relationship of data bias and model bias below.

### 4.4 Impact of Training Data

We hypothesise that training data is an important factor for erasure: models underpredict countries which appear in the data infrequently compared to their population. To study the relationship between training data bias and model bias, we extract the distribution of country mentions in the training data. We consider the Pile dataset (Gao et al., 2021) used to pre-train the GPT-Neo LLMs analysed in this study. To determine the probability of occur-

rence in the training data $p^{\text{train}}(x)$, we compute the number of times each country $x$ is mentioned in the dataset, i.e., $p^{\text{train}}(x) \propto$ *# mentions of x*. These mention counts are weighted by the number of training epochs this document was included while training (dataset weights $w_d$ from Gao et al. (2021)). We account for alternative country names as described in §3.1. Thus, the final formula becomes

$$p^{\text{train}}(x) \propto \sum_{d \in \text{datasets}} w_d \sum_{a \in \mathcal{A}} \#x^a \in d, \qquad (11)$$

where $\mathcal{A}$ represents a set of alternative country names of a given country and # represents counts. Once all the counts are gathered, the results are normalised to determine the final values of $p^{\text{train}}(x)$, which we compare to the outputs of LLMs.

Specifically, we compare ground truth $p^{\text{true}}(x)$, training data $p^{\text{train}}(x)$ and GPT-NeoX predictions $p(x)$ for countries $x \in \mathcal{S}_3$ (Figure 4, right). We see that countries experiencing erasure are indeed underrepresented in the training data, and the prediction probabilities of these countries are similar to their frequency distribution in the training corpus ($p^{\text{true}}(x) \gg p(x) \approx p^{\text{train}}(x)$).

We then compute erasure against the training data, $ER^r(p^{\text{train}}, p)$, i.e., considering the ground truth to be $p^{\text{train}}$ (Figure 4, middle). We find that erasure values in this case are considerably lower. For

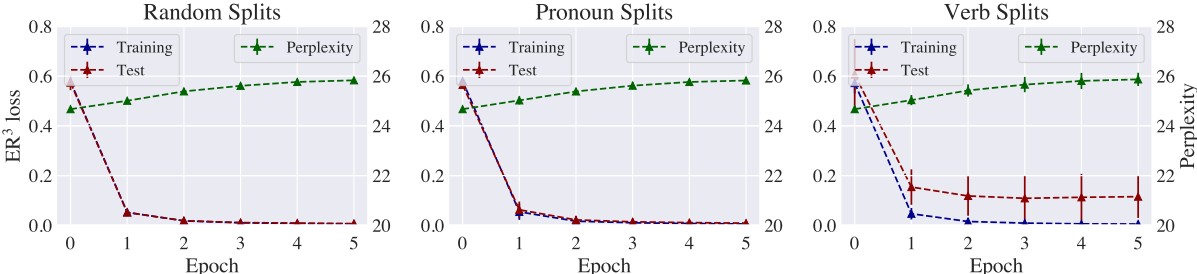

Figure 5: **Finetuning effectively alleviates erasure.** We plot average $ER^r$ on training (blue) and test (red) set prompts during 5 epochs of finetuning of the GPT2-small model. Error bars indicate minima/maxima over 5 folds.

instance $ER^r(p^{\text{train}}, p^{\text{agg\_model}}) = 0.08$ for GPT-NeoX compared to $ER^r(p^{\text{true}}, p^{\text{agg\_model}}) = 0.46$, i.e., erasure using the world population (Figure 4, left). This indicates that the GPTNeoX family of LLMs mimic the training distribution (of country mentions). Furthermore, we find that the erasure score of the training data compared to ground truth, $ER^r(p^{\text{true}}, p^{\text{train}})$, is itself $0.46$, which closely matches the erasure for models trained on this data. The high correlation between data bias and model bias suggests the composition of training data is a key source of erasure in the investigated LLMs.

## 4.5 Mitigation

In this section, we explore finetuning as a strategy to mitigate erasure. We perform gradient updates on a pretrained GPT2 model to minimise the erasure loss $ER^3(p^{\text{true}}, p)$ on the training data given by prompt data set $\mathcal{D}$. We note that our finetuning strategy differs from the related approach from §6.2 of Zhou et al. (2022) in that we have formulated a loss function which allows us to perform supervised finetuning. Zhou et al. (2022) instead continue training the model using the standard masked language modelling loss with augmented data related to underpredicted countries.

We use the AdamW (Loshchilov and Hutter, 2017) optimiser with learning rate $3e - 5$ and train for an additional 5 epochs (including one epoch of warmup under a linear schedule). We find that due to our loss function's direct dependency on the logits and the re-normalisation of probabilities over $\mathcal{X}$ (2) our finetuning strategy works best for deterministic models, hence we set dropout rates to 0 for embeddings, encoder, pooling and attention layers. For finetuning, we update the bias terms only, following Zaken et al. (2021). This is a memory efficient strategy that is expected to work particularly well in settings with constant outputs (we want the

generated distribution for *all* our prompts to match the ground truth), while not impacting the general language modelling abilities. We evaluate whether the language modelling abilities deteriorate by measuring perplexity on Wikitext-2-v1 (Merity et al., 2016) before and after every epoch of finetuning.

To measure how well our finetuning strategy generalises, we compare three different ways of performing train-test splits of our 955 prompts. These include *Random* partitioning: we randomly split the prompts into $75\%$ training and $25\%$ test data; *Pronouns*: we split the prompts based on the pronouns they contain, e.g. all prompts containing "she", "you", "we" and "they" are in the training set, "I" and "'he' in the test set; *Verbs*: we divide along verb groups, e.g. prompts containing "to live in" and 'to be a citizen' of are in the training set, "to reside in" is in the test set. These three setups require increasing levels of generalisation.

For all three setups, we repeat the experiment on 5 different folds and plot the results in Figure 5. We find that our finetuning strategy is effective: the average erasure $\frac{1}{|\mathcal{D}|} \sum_{c \in \mathcal{D}} ER^3(p^{\text{true}}, p(\cdot|c))$ is small after 5 epochs of finetuning, both on training (blue) and test data (red). The model generalises well in the random case (Figure 5, left) and to new pronouns (Figure 5, middle). As expected, *verb splits* are the most challenging for our model, where we see that the erasure values decrease but not as much as we see in other splits (Figure 5, right). In all cases, we see only a small deterioration in language modelling performance, as indicated by an approximate $5\%$ increase in the perplexity (The green lines in the plots of Figure 5 correspond to perplexity). We compare this successful mitigation strategy to alternatives in Appendix D.

# 5 Conclusion

We motivated the need for large language models to be more geographically inclusive—which remains to be an overlooked aspect of inclusive model development. Specifically, we studied and formalised a notion of geographical erasure, which captures the countries that are underpredicted and the extent to which they are underpredicted. We discussed how our formulation captures many desirable properties. In our experiments, we found clear instances of geographical erasure, which were consistently observed across 10 different language models. Perhaps unsurprisingly, the output probabilities of language models closely follow the frequencies of country mentions in the training corpora, a likely cause of erasure. We examined a finetuning-based mitigation strategy and found it to be effective in alleviating erasure.

# 6 Limitations

**Languages considered.** We limit our analysis to models trained on English texts, and hence we prompt them in English only. Our methodology extends to other languages straightforwardly. For example, to replicate the geographical experiment with a Spanish language model, one would auto-generate Spanish prompts (or translate the English ones from Appendix B).

The language (of prompts) used to analyze erasure should be accounted for while collecting ground truth data: for instance, English speaking countries are expected to have higher probability conditioned on "I live in", and similarly Spanish speaking countries conditioned on "Vivo en" are likely to have higher probabilities. In our work, we factor this by considering English speaking populations as ground truth in §4 (and one would proceed accordingly for a model in a different language).

**Difficulty in obtaining ground truth.** Language specific ground truth data is less reliable and harder to obtain than raw population counts. Such statistics are often self-reported and the level of proficiency differs dramatically across regions, especially since the numbers include second language speakers.[2] Since we only measure erasure for countries where $p_i^{\text{true}}$ is available, the availability of language specific ground truth data is itself a biasing factor. This is evident from Figure 1, which depicts how the lack of ground truth data predominantly affects central African regions.

**Knowledge encoding vs. language generation.** Our erasure metric is based on country probabilities given a prompt $p(x_i|c)$. These probabilities can be interpreted as knowledge encoded by the model. When generating text, the model probabilities are used to sample next tokens. Sampling (or decoding) can be performed using different strategies, e.g. greedy or beam search (Klein et al., 2017) to maximise probabilities, or top-k (Fan et al., 2018) and top-p (Holtzman et al., 2019) sampling strategies to generate more diverse outputs.

Our work does not analyse the effect of the decoding mechanism since we work directly on $p(x_i|c)$ instead of generated text. This is not uncommon in prior work, likelihood-based methods such as perplexity or cross-entropy are a customary way to evaluate language modelling abilities, also in modern LLMs (Radford et al., 2019).

Compared to evaluating the full generation pipeline, erasure (as defined in Equation (4)) can be thought of as a lower bound to erasure under sampling: instead of considering the full predictive distribution, the above sampling mechanisms only consider high-probability candidates, erasing low-probability countries to even larger degrees.

**Causes of erasure.** Our analysis covers two potential sources for erasure: training data and model size. Model bias is commonly explained by data bias (e.g. Bender et al. (2021); Schwöbel (2022) and Buolamwini and Gebru (2018)). In our work, we have not experimentally established the cause, our experiments instead indicate a high correlation of model biases and data biases in §4.4, suggesting that data is a likely source of erasure. Data, however, is not the only biasing factor. Model architecture and training paradigm determine how the data is used by the model. Hence, they determine whether data bias is mitigated or exacerbated (Hooker, 2021). We examine the impact of model size and find that it has little to no impact on geographical erasure ( §4.3)). Examining the impact of other factors on erasure is left to future work.

# Acknowledgements

This work greatly benefited from discussions with Mansi Gupta, Luca Franceschi, Bilal Zafar, Gianluca Detommaso, Martin Wistuba and Prabhu Teja Sivaprasad. We also appreciate the feedback on our work by Navreet Kaur and anonymous EMNLP reviewers. DP acknowledges Adobe Inc. and Kotak IISc AI-ML Centre (KIAC) for their support.

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

## A Choosing $r$ – additional Models

Section 3.4 compares $\text{ER}^r$ for different values of $r$ to the KL-divergence. We pick $r = 3$ in this experiment such that $\text{ER}(p^{\text{true}}, p)^r \approx \text{KL}(p^{\text{true}}||p)$. Figure 6 contains the same experiment for all models under consideration. The optimal choice according to this heuristic is $r = 3$ for all of them.

## B Details on Prompt Rephrasing

As described in §3.5, we formulate different versions of a seed prompt $c =$ "I am from ___." in order to produce a set of diverse prompt wordings $\mathcal{D}$. We do this in two steps. First, we prompt Chat-GPT[8] to rephrase the seed prompt "I am from". We manually remove irrelevant prompts, e.g.. those containing country adjectives rather than nouns. This first step extends the seed prompt into to the following 16 base formulations.

### B.1 List of prompts

1. I am from ___.

2. I live in ___.

3. I hail from ___.

4. My homeland is ___.

5. I come from ___.

6. I was born and raised in ___.

7. I am a citizen of ___.

8. I originate from ___.

9. My roots are in ___.

10. I grew up in ___.

11. I was brought up in ___.

12. I was raised in ___.

13. I was born in ___.

14. My place of origin is ___.

15. I reside in ___.

16. My home country is ___.

---

[8]Accessed via https://chat.openai.com/.

Secondly, we apply template-based rephrasing similar to Romano et al. (2006). We replace the subject of the sentence from "I" to alternative pronouns {"She", "He", "We", "They", "You"} – adjusting verb forms accordingly – as well as combinations of {"My", "Her", "His", "Our", "Their"} × { "uncle", "aunt", "brother", "sister", "niece", "nephew", "mother", "father", "mom", "daughter", "son", "cousin", "friend", "relative"}. This second step extends the 16 base formulations into our final dataset $\mathcal{D}$ containing 955 prompts.

## C Variability across Prompts

Recall from (8) that

$$p_i^{\text{agg\_uni}} = p(x_i|\mathcal{M}) = \mathbb{E}_{p(c|\mathcal{M})}\left[p(x_i|c)\right]$$

under uniform $p(c|\mathcal{M})$. By Jensen's inequality,

$$\text{ER}^r(p^{\text{true}}, p^{\text{agg\_uni}}) =$$
$$\sum_{i \in \mathcal{S}_r} p_i^{\text{true}} \log\left(\frac{p_i^{\text{true}}}{\mathbb{E}_{p(c|\mathcal{M})}\left[p(x_i|c)\right]}\right)$$
$$\leq \mathbb{E}_{p(c|\mathcal{M})}\left[\sum_{i \in \mathcal{S}_r} p_i^{\text{true}} \log\left(\frac{p_i^{\text{true}}}{p(x_i|c)}\right)\right]$$
$$= \frac{1}{C} \sum_{c \in \mathcal{D}} \text{ER}^r(p_i^{\text{true}}, p(x_i|c)).$$

Thus, erasure under the aggregate distribution $\text{ER}^r(p^{\text{true}}, p^{\text{agg\_uni}})$ is a lower bound to the average erasure.

## D Alternative Mitigation Strategies

**Finetuning for other values of $r$:** In §4.5 we mitigate erasure by finetuning, employing $\text{ER}^3$ as a loss function (Figure 5). This choice corresponds to a minimal intervention where we only modify the distributions for affected countries at a rate above $r = 3$; we do not address any underprediction by a smaller degree.

Finetuning with $r = 0$ (Fig. 7) is a stronger intervention, matching the distributions for *all* countries (since $\text{EB}^0(p^{\text{true}}, p) = \text{KL}(p^{\text{true}}||p)$, see §3.4). As before, we can match the full distributions and achieve $\text{EB}^0(p^{\text{true}}, p) \approx 0$ after only 5 epochs of finetuning. However, due to the more drastic intervention into the model distribution $p^{\text{true}}$ the drop in language modelling performance is larger. Perplexity increases by almost 20% compared to 5% in Figure 5. Note the different y-axis scales between Figures 5 and 7.

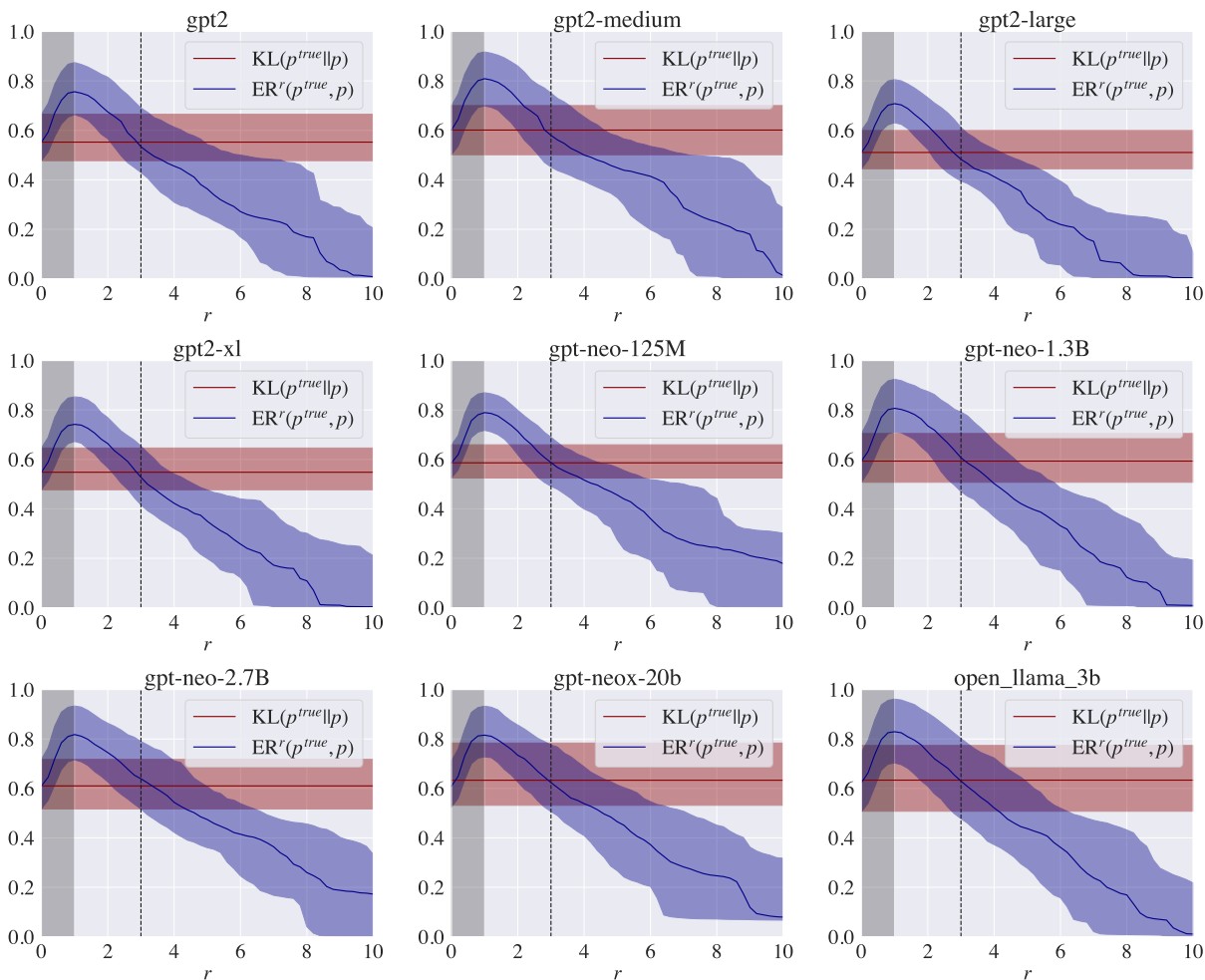

Figure 6: **The relationship between $ER^r$ and KL-divergence is similar for all models.** We compare $ER^r$ (blue) for different $r$ to the KL-divergence (red), repeating the experiment from §3.4 for all models. The median is plotted as a solid line, $25^{th} - 75^{th}$ percentiles as shaded areas (over different versions of the "I live in" -prompt, see §3.5). We pick $r = 3$, the integer value for which KL and $ER^r$ are the most similar for all models (dashed line, black).

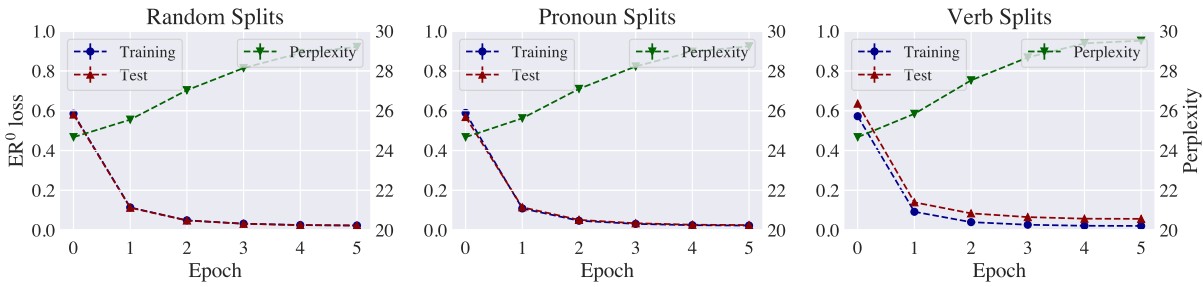

Figure 7: **Finetuning at $r = 0$ also mitigates erasure, though at a higher perplexity cost.** Like in Figure 5 we plot average $ER^r$ on training (blue) and test (red) set prompts during 5 epochs of finetuning of the GPT2-small model. Error bars indicate minima/maxima over 5 folds. Note the different y-axis scale compared to Figure 7.

| | No mitigation | $ER^3$ | $ER^0$ | $\tau$ |
|---|---|---|---|---|
| Training loss | $0.5722 \pm 0.0160$ | $\mathbf{0.0054 \pm 0.0003}$ | $0.0211 \pm 0.0003$ | $0.5641$ |
| Test loss | $0.6065 \pm 0.0543$ | $\mathbf{0.0068 \pm 0.0014}$ | $0.0566 \pm 0.0020$ | $-$ |
| Perplexity | $24.6716 \pm 0$ | $25.8694 \pm 0.0913$ | $29.5201 \pm 0.0297$ | $\mathbf{24.5357}$ |

Table 1: **Summary of mitigation experiments.** Finetuning with $ER^3$, $ER^0$ and mitigation via optimising $\tau$. We report the training and test loss as well as perplexity after the $5^{th}$ finetuning epoch. Best values bolded.

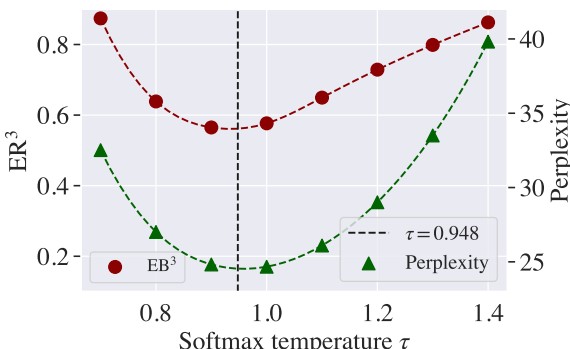

Figure 8: **Mitigating ER$^r$ using the temperature parameter $\tau$ is less successful than full finetuning.** ER$^r$ and perplexity are plotted as a function of $\tau$. The optimal value (minimising ER$^r$ w.r.t. $\tau$) is 0.948, dashed line.

**Mitigation via Temperature Softmax**: A simple way to modify the model distribution $p$ is via the softmax temperature parameter $\tau$ of the model. We have used $\tau = 1$ in all previous experiments. Here, we experiment with modifying $\tau$ to mitigate ER$^r(p^{\text{true}}, p)$ such that

$$\text{ER}^r = \min_{\tau} \text{ER}^r(p^{\text{true}}, p_\tau). \qquad (12)$$

Figure 8 shows ER$^r$ and perplexity as a function of $\tau$. The optimal value (minimising ER$^r$ w.r.t. $\tau$) is 0.948, dashed line. This mitigation method is compared to fine-tuning of the neural network parameters from earlier experiments in Table 1. The two middle columns correspond to the finetuning results from Figure 5 and Figure 7, the rightmost column contains the results for varying temperature parameter $\tau$.

Perhaps unsurprisingly, mitigation attempts with a single parameter $\tau$ are much less successful than using full finetuning (small drop in ER$^r$ only, see first row of Table 1). Perplexity, however, improves slightly over the original model.