# OpenReview forum: "Geographical Erasure in Language Generation"
_EMNLP/2023/Conference — EMNLP 2023 Findings_

### Official Review · Reviewer_u9cg · 2023-08-04

**Soundness:** 4

**Excitement:**

4: Strong: This paper deepens the understanding of some phenomenon or lowers the barriers to an existing research direction.

**Missing References:**

[1] Hooker, Sara. "Moving beyond “algorithmic bias is a data problem”." Patterns 2.4 (2021).

[2] Zylberajch, Hugo, Piyawat Lertvittayakumjorn, and Francesca Toni. "HILDIF: Interactive debugging of NLI models using influence functions." Proceedings of the First Workshop on Interactive Learning for Natural Language Processing. 2021.

Some more references on model size and biases (so far there seems to be no clear conclusion wrt. this matter):

* Tal, Yarden, Inbal Magar, and Roy Schwartz. "Fewer Errors, but More Stereotypes? The Effect of Model Size on Gender Bias." Proceedings of the 4th Workshop on Gender Bias in Natural Language Processing (GeBNLP). 2022.
* Hooker, S., Moorosi, N., Clark, G., Bengio, S., & Denton, E. (2020). Characterising bias in compressed models. arXiv preprint arXiv:2010.03058.
* Silva, Andrew, Pradyumna Tambwekar, and Matthew Gombolay. "Towards a comprehensive understanding and accurate evaluation of societal biases in pre-trained transformers." Proceedings of the 2021 Conference of the North American Chapter of the Association for Computational Linguistics: Human Language Technologies. 2021.
* Bender, E. M., Gebru, T., McMillan-Major, A., & Shmitchell, S. (2021, March). On the dangers of stochastic parrots: Can language models be too big?🦜. In Proceedings of the 2021 ACM conference on fairness, accountability, and transparency (pp. 610-623).

**Paper Topic And Main Contributions:**

This work presents a novel bias mitigation method focused on mitigating geographic biases of LLM generated texts. To do so, the authors define erasure as the 'KL divergence' between true and predicted distributions for 'erased' sets that are constructed using a user-defined threshold r. They then continue to construct 16 different prompts for probing LLMs. In their experiments, the authors probe four different LLMs (with overall ten different model sizes). While they find that a majority of countries (~82.7%) do not suffer from erasure for r=3, they also identify 10 countries that are 'erased' across all models. Their additional analysis then identifies the training data (the Pile) a likely source of such erasure. Finally, the authors show that utilizing their metric for fine-tuning can mitigate existing geographic biases with small deterioration in model performance.

**Questions For The Authors:**

* Could you elaborate on why one could not simply try to mitigate the bias by fine-tuning the model to express the true distribution?

* Source of erasure: A start would be to replace/shuffle country mentions and re-train models (possibly on a smaller scale) and check if they display similar behaviors. An alternative could be methods that use influence functions [2].  (Note: This is not a request for additional experiments, but a suggestion for future works)

**Reasons To Accept:**

This work presents a very interesting study on geographic bias in LLMs. The authors propose a sound method, an interesting analysis, and show the usefulness of their approach across different models (and sizes). Also, they promise to release their code upon publication.

**Reasons To Reject:**

Overall, the presented work is interesting and presents various analysis across several models. Some points that could be addressed to improve the paper are:

* While the proposed method is easy to apply directly during fine-tuning (as a loss), one limitation is the transferability to other types of biases where it is more difficult to obtain ground-truth distributions or an exhaustive list of targets to mitigate. (should be included in the limitations section)

* The claim that the training data is a key source for geographical erasure is very strong and should be hedged. While the analysis that reveals a high correlation indicates that this is very likely, additional experiments would be necessary to deduce causation. (See suggestions and also [1])

**Reproducibility:**

5: Could easily reproduce the results.

**Reviewer Confidence:**

3: Pretty sure, but there's a chance I missed something. Although I have a good feel for this area in general, I did not carefully check the paper's details, e.g., the math, experimental design, or novelty.

---

> ### Author Rebuttal · Authors · 2023-08-28
>
> We thank the reviewer for their careful analysis and insightful feedback. We are glad to learn that they found our analysis to be interesting, methods to be sound, and our proposal of measuring and reducing erasure to be useful. We are also grateful for the references and suggestions to improve our work.
>
> __Ground truth and transferability [Reasons to reject]__
>
> Our proposed measure of erasure, and subsequent mitigation, requires a reference distribution (ground truth). This reference should capture some notion of ideal or aspired model behaviour. We acknowledge the difficulty of obtaining ground truth for the geography case in §6: Languages Considered. Our methodology could be applied to other domains and other types of biases. For new domains it may be more difficult to obtain ground truth. On the other hand, there might be domains for which it is easier (e.g., cases where a uniform distribution is appropriate as is assumed in other works, e.g. Liang et al., ‘22). We will expand our discussion on this in the limitation section.
>
> More generally, it is difficult to effectively measure and tackle biases for domains where ground truth data or a notion of ideal model behaviour is not available. For example, even simple fairness notions like demographic parity can only be computed if the demographic group label is available (gender, race, etc.).  This is a shortcoming of supervised bias mitigation strategies rather than our method specifically. We will include this discussion in the limitations of our method.
>
>
> __Sources of erasure: causation vs. correlation [Reasons to reject, Q2]__
>
> Our claim of causation might indeed be overly strong. We will update the text to state high correlation rather than causation.
>
> We appreciate your suggestions to experimentally support claims of causation. We hypothesise that after performing the intervention you propose (replacing/shuffling country mentions to obtain uniform rates during training), we would see little-to-no erasure.
> Similarly, the suggestion about using influence functions to verify causes is also an interesting one. As you mention, retraining a model and computing influence scores to establish causation is outside the scope of the current submission. We are looking forward to exploring these directions in future work.
>
> __Fine-tuning the model to express the true distribution [Q1]__
>
> This is a great idea and our methodology supports it. Specifically, if we set $r=0$ we include all the terms in the KL-divergence (see §3.4: The choice of $r$). Hence, we minimise the KL-divergence between $p^{true}$ and $p$ on the full domain, i.e, all countries.
>
> The reason why we did not include this in the original draft is as follows: If one indeed assumes that erasure up to a factor $r$ is acceptable (a user choice), then finetuning with respect to the chosen $\text{ER}^r$ is a minimal intervention in the sense that it will only modify the distributions for the affected countries at rate $r$. As such, it has a relatively small impact on model perplexity (see Fig. 6 for $r=3$). Since more countries are affected, we would expect finetuning on the full distribution ($r=0$) to have a much bigger effect on perplexity.
>
> We ran the experiment and indeed found that the impact on perplexity is larger when we train with $r=0$ than with $r=3$; perplexity increases by almost 20% compared to 5%. See this [figure](https://www.dropbox.com/scl/fi/ac9l3xak9kp8r6dfegiu9/finetuning_r-0.png?rlkey=4v6q1fdbasxh3ue75tboufzcp&dl=0), and note the different y-axis label compared to Fig. 6 in the draft.
>
> ___
>
> We believe that the additional experiment and explanations provide “extra support or details” to address the reviewer’s soundness concerns. We hope the reviewer will consider increasing their soundness score, or engage with us in further discussion to point out points that might remain missing.

---

### Official Review · Reviewer_YS6F · 2023-08-05

**Soundness:** 4

**Excitement:**

4: Strong: This paper deepens the understanding of some phenomenon or lowers the barriers to an existing research direction.

**Paper Topic And Main Contributions:**

This paper aims to address the unbalance and bias happened in pre-training of large language models, specifically for the geographical erasure situation, which is lack of focus. The paper demonstrates consistent instances of erasure across a range of LLMs, of which results can be largely explained by the frequencies of country mentions in training corpora. Although, this paper propose a custom objective for fune-tuning to mitigate the geographical erasure.

**Questions For The Authors:**

Although a series of figures with description could illustrate the experimental settings and design, a clear table with some case study could further help readers to understand the paper, like how the prompt sentences work. Besides, a precise results of experiments listed in the table (like add in the appendix) could bring more inspiration.

**Reasons To Accept:**

1. The paper focuses on a relative unpopular objective, geographical erasure, rather than common discussion on representational harms such as biases related to race, gender, occupation and sexual orientation. However, a complete reading of the problem is given and a series of related experiments are designed on many common large language models to prove the conclusions.

2. This paper proposes a measurement ER to measure the erasure. And base on that the analyses for the models' ability on geographical representation are discussed. Beyond that, the author further proposes a fine-tuning strategy by optimizing the ER to mitigate the erasure problem. The whole framework of this study is a full closure of the loop.

3. Several figures in the paper are well designed and clearly demonstrate the relationship between the pre-training corpora and the model's erasure capability.

**Reasons To Reject:**

The presentation of ER measurement is mostly based on frequency and probability, which is intuitive however less interpretability. More discussion regarding ER could be elaborated.

**Reproducibility:**

4: Could mostly reproduce the results, but there may be some variation because of sample variance or minor variations in their interpretation of the protocol or method.

**Reviewer Confidence:**

3: Pretty sure, but there's a chance I missed something. Although I have a good feel for this area in general, I did not carefully check the paper's details, e.g., the math, experimental design, or novelty.

---

> ### Author Rebuttal · Authors · 2023-08-28
>
> We thank the reviewer for their feedback and endorsement of our work.
>
> __Interpretability of $\text{ER}^r$:__
>
> Section 3.4 and Fig. 2 aim to give some theoretical and empirical guidance on how to interpret $\text{ER}^r$. We appreciate the callout for more discussion on the interpretability of the ER metric and will include additional details in the revision. If the reviewer has any concrete questions or experimental suggestions we will make sure to integrate them.
>
> __Table with experimental detail and reproducibility:__
>
> Thanks for the suggestion, we will happily add a table containing the results of our experiments numerically, along with additional details of the experimental setup. We will also release the code upon publication (see footnote 1). In combination, we hope that these two points will address the difficulty the reviewer sees in terms of reproducibility.

---

### Official Review · Reviewer_U4e2 · 2023-08-11

**Soundness:** 3

**Excitement:**

2: Mediocre: This paper makes marginal contributions (vs non-contemporaneous work), so I would rather not see it in the conference.

**Paper Topic And Main Contributions:**

This paper focuses on measuring and removing the models’ bias towards different countries. They first propose a metric and then use it to measure the importance of different countries to the models. Finally, they reduce the models’ bias by fintuning.

**Questions For The Authors:**

Why do we need to make the probabilities of ‘I am from ___.' the same as the portion of people who speak English? Moreover, someone who can speak English does not necessarily mean that he/she will only use English on the internet. For example, numerous Japanese people can speak English but that does not mean that they will type English on the Internet.

**Reasons To Accept:**

1. This paper focus on identifying and reduce the problem of underrepresented countries by models, which seems to be novel.
2. This paper further utilizes the loss function proposed by Zhou et al. (2022) to mitigate the underrepresented phenomenon.

**Reasons To Reject:**

1.  The loss objective is not novel.
2. This problem is not novel.
3. The prompts seem to be not reasonable as it does not have any other contexts. Evaluating the relationship between English Speaking Population and Prediction Probabilities is not reasonable as not everyone uses the internet and not everyone has access to the internet.
4. Moreover, it is not sure whether English Speaking Population contains international students or workers.
5. It is still unknown why models need to treat all countries the same as the objective of pretraining or finetuning is to make the model predict test data. And in test data, that kind of unbiased thing is natural.

**Reproducibility:**

5: Could easily reproduce the results.

**Reviewer Confidence:**

3: Pretty sure, but there's a chance I missed something. Although I have a good feel for this area in general, I did not carefully check the paper's details, e.g., the math, experimental design, or novelty.

---

> ### Author Rebuttal · Authors · 2023-08-28
>
> We thank the reviewer for their comments. We’d like to clarify the following points:
>
> **Novelty of our work [as mentioned in reasons to reject]:**
>
> The literature on fairness and biases has predominantly focused on concerns related to gender, race, occupation, and (in some cases) sexual orientation. For perspective, there are at least 304 published papers that analyse gender bias in NLP (as per Stanczak & Augenstein, 2021). Comparatively, there are few studies that analyse models for geographical biases, and as the reviewer notes, Zhou et al. (2022) is an important exception. The work from Zhou et al. is related to ours in spirit, as it also aims to study biases from a geographical lens. Their work primarily concerns the quality of embeddings for different nations, and they briefly hint at how certain richer countries might be over-predicted. Our work attempts to understand the degree to which certain countries may be underpredicted in generation: to do so, we formalize a notion of geographical erasure, measure erasure for different models (with varying sizes), identify the potential sources of erasure, and evaluate techniques to mitigate it.
>
> > The loss objective is not novel.
>
> This is _not_ true: Unlike us, Zhou et al. do not propose a custom metric that can be used as a loss for fine-tuning. Their analysis is based on empirical distributions (i.e., the frequency with which country names appear in training data and model generations). In §6.2 they finetune on modified training data using the cross entropy loss. We measure erasure directly using the distributions parameterised by the model which gives rise to a novel, effective finetuning objective.
>
> **Reference Distributions**
>
> > “why models need to treat all countries the same as the objective of pretraining” and “Why do we need to make the probabilities of ‘I am from ___.' the same as the portion of people who speak English?”
>
> Our proposed measure of erasure, and subsequent mitigation, requires a reference distribution (ground truth). This reference should capture some notion of ideal (or aspired) model behaviour. Measuring or mitigating bias is inherently a normative process. We chose a reference distribution of the English speaking population, since the investigated models are monolingual (trained and used in English language). This reference distribution takes inspiration from principles of proportional representation (a notion common in electoral systems), wherein the benefits are divided in the same proportions as the number of people in a given demography. Having said this, one could directly use our formulation for a different reference distribution.

---

### Official Review · Reviewer_MJ7R · 2023-08-12

**Soundness:** 4

**Excitement:**

4: Strong: This paper deepens the understanding of some phenomenon or lowers the barriers to an existing research direction.

**Paper Topic And Main Contributions:**

The paper quantitatively explores the underprediction of certain countries by large language models in language generation. The authors term this as "geographical erasure" and propose a measure to calculate the extent to which some LLMs can be affected by geographical erasure. The key analytical findings of the paper include that many autoregressive LLMs suffer from geographical erasure; model size is uncorrelated with the measure proposed, indicating all models, big and small, may suffer from geographical erasure; and erasure of countries is directly proportional to the frequency of the country mentions in the training data. Finally, supervised finetuning is shown to mitigate erasure without adversely impacting LLMs generative performance.

**Questions For The Authors:**

Q1. On line 177, what does the subscript c stand for? Similar in 186. It's confusing that you define the variable and then use it a subscript.

Q2. From lines 206-207: The ground-truth probabilities are computed from a wikipedia list but the link only lists 12 countries. Why is that? The population statistics is also quite suspect: for example, the size of the English speaking population in the US is the same as the population of the country. Since this is a critical piece in the measurement of erasure, the authors should be more transparent of how they calculated the groundtruth probabilities.

* A stylistic comment: lines 212-216 aren't related to ground truth and it would improve the readability if it is folded in section 3.1

Q3. The summands in Eq.4 can be written as \sum_i p_i^{true}\logp_i^{true} - \sum_i p_i^{true}\log p_i. Unless I'm mistaken, the left part (the entropy) is model independent and will remain the same for any model. Why not just compute the right part (the cross entropy) as the erasure metric?

Q4. I wouldn't characterize the analysis in 4.4 as "causes for erasure". The paper shows quite clearly that there is correlation between training frequency and the measure of bias that is proposed. Other research has also linked training frequency with bias or accuracy in generation. However, the evidence is correlation and not causation, isn't it?  Training frequency is an artifact of the data and the training protocol and not the (sole) reason for the bias; conversely, if the frequency of country mentions was controlled during training to be equal, it's still likely that you would see erasure. Could the authors provide more explanation?

Q5. The model probabilities could be wildly different if, say for example, the temperature parameter is varied. How should the measure be adjusted to account for the stochasticity due to such hyper parameters? Is that a problem that the authors have encountered?

**Reasons To Accept:**

The proposed measure for erasure has many desirable properties such as interpretability, ease in computation, and generalizability beyond geographical erasure to other types of erasure. The application of this measure for other attributes could help us understand the range of the erasure biases LLMs hold and take steps to mitigate them. The proposed measure also lends itself nicely to accounting for erasure in a loss function and finetuning the model specifically to alleviate erasure.

**Reasons To Reject:**

Even though I think the analyses are generally sound and very illuminating, I found some inconsistency in how the groundtruth probabilities were constructed (please see Q2) . Since it's a crucial piece in the measure proposed, it can crucially affect the downstream analysis and mitigation. It is also unclear if and how the measure should be adjusted to account for stochasticity in generation, such as the use of temperature parameter to control generation.

**Reproducibility:**

4: Could mostly reproduce the results, but there may be some variation because of sample variance or minor variations in their interpretation of the protocol or method.

**Reviewer Confidence:**

4: Quite sure. I tried to check the important points carefully. It's unlikely, though conceivable, that I missed something that should affect my ratings.

**Typos Grammar Style And Presentation Improvements:**

* On line 55, need a space at the start of the sentence
* On line 355, plots --> prompts
* On line 392, one --> on

---

> ### Author Rebuttal · Authors · 2023-08-28
>
> We thank the reviewer for their in-depth review and are glad to learn that they found our study sound and illuminating, and appreciated the desirable properties of the erasure measure. We will gladly incorporate all suggestions, including clarification on correlation vs. causation as well as presentation and stylistic improvements. Please find detailed comments below.
>
>
> __Subscript of x [Q1]__
>
> This is a slight abuse of notation which was meant to improve readability. The subscript $c$ of $x$ indicates that $x$ is a token appearing in the prompt $c$. The superscript indicates the position. E.g., for prompt c=”I live in” we write $x_c^1 = $“I”, $x_c^2=$“live”, and so forth. We will clarify this in our draft.
>
>
> __Wikipedia list used for ground truth [Reasons to reject & Q2]__
>
> > Q2 (i): From lines 206-207: The ground-truth probabilities are computed from a Wikipedia list but the link only lists 12 countries. Why is that?
>
> The discussed [Wikipedia list](https://en.wikipedia.org/wiki/List_of_countries_by_English-speaking_population) contains $127$ countries (at the time of writing the paper and as on August 26, 2023). Those are the $M=127$ countries we include in our analysis (see Sec. 3.1; will be moved to 3.2 as per stylistic comment). We believe the confusion might arise from the fact that we only “show the 12 countries with the largest English speaking populations” (see Figure 2, caption). We will update the section to clarify this.
>
> > Q2 (ii): The population statistics is also quite suspect: for example, the size of the English speaking population in the US is the same as the population of the country.
>
> For most countries, the data comes from official census reports, which have been collated in the Wikipedia page on [List of countries by English-speaking population](https://en.wikipedia.org/wiki/List_of_countries_by_English-speaking_population). We directly use the data summarised in the Wikipedia page. For the US, figures come from the 2011–2015 American Community Survey 5-year estimates by the U.S. Census Bureau for persons age 5 and older. Total English speakers are those who either spoke English at home (i.e. as first language), or reported speaking another language at home but could speak English "very well" or "well" (i.e. as an additional language). The total population as per the survey was about 331 million, of which 316 million people were English speakers (about 95% of the population).
>
>
> __Cross-entropy vs. KL-divergence [Q3]__
>
> Yes, one can write
> $ \text{KL}(p^{true} || p) = \sum_i p_i^{true} \log p_i^{true} - \sum_i p_i^{true}\log p_i = - H(p^{true}) + H(p^{true}, p), $
>
> where the entropy term $H(p^{true})$ is constant.
>
> We include the constant term in our definition of $\text{ER}^r$ for the sake of interpretability: Consider the case $r=0$. If $p^{true}=p$ we would like to obtain that $\text{ER}^0=0$, because no erasure occurs when the distributions match. If we instead defined erasure as the cross-entropy $\text{ER}^r := H(p^{true}, p)$, we would obtain non-zero erasure for matching distributions (we would then have $\text{ER}^0(p^{true}, p^{true})=H(p^{true})$).
>
>
> __Training data: causation vs. correlation [Q4]__
>
> Thanks for pointing this out, it is true that we have only shown correlation and not causation between training data bias and model bias. We will update the paper to make sure that any hypothesis we propose is clearly marked as such and cannot be confused with verified claims.
>
> We much appreciate your experiment suggestion of controlling “the frequency of country mentions [...] during training to be equal”. Together with the related suggestions by reviewer u9cg, we will consider this in future work.
>
>
> __Temperature parameter and stochasticity during generation [Reasons to reject & Q5]__
>
> We analyse the probability distribution that a model assigns to the candidate tokens. Specifically, we apply softmax to the logits returned by the language modelling head to obtain $p(x_k | c)$. Applying a temperature parameter $\tau$ to this softmax directly modifies $p_{\tau}(x_k | c)$. Hence, it indeed has a very direct effect on $\text{ER}^r$. In our work, we have treated this parameter as fixed ($\tau=1$), but the measure could be adjusted to account for a free temperature parameter $p_{\tau}(x_k | c)$ by defining $\text{ER}^r$ as the minimum over $\tau$:
>
> $\text{ER}^r = \min_{\tau} \text{ER}^r( p^{true}, p_{\tau})$.
>
> Picking the optimal tau in this way can be viewed as a single-parameter erasure mitigation strategy: we modify $p_{\tau}(x_k | c)$, until we find the parameter that minimises erasure. We hypothesise that this strategy will be somewhat successful at mitigating erasure, but will come at the cost of a large perplexity increase. Smaller temperatures lead to predicting the low probability countries more often but will likely present rather drastic interventions in the model distribution (similar, but even more extreme than the $r=0$ experiment we added under review u9cg). We will add this experiment to the camera ready version.
>
> For sampling strategies (beam search, top-k, top-p, etc.) and their impact on erasure see §6: Knowledge encoding vs. language generation.
>
> ___
>
> We believe that we address the reviewer’s concerns and have clarified “minor points [that] may need extra support or details” and sincerely hope that the reviewer will consider increasing their soundness rating.

---

### Meta-Review · Area_Chair_vFeb · 2023-09-19

**Recommendation:** 3

**Metareview:**

The paper "Geographical Erasure in Language Generation" presents work on the imbalance of generated data from LLMs, more concrete about the lack of representation of certain languages in generated data. The paper also presents a method to counter this phenomenon.

Arguments for rejecting the paper mentioned by the reviewers are related to the novelty of the task and details on basic assumptions.
Arguments for accepting the paper mentioned by the reviewers are related to the aspect studied (geographic erasure) and its potential to extend this to other similar phenomena.

---

### Decision · Program_Chairs · 2023-10-07

**Decision:**

Accept-Findings

**Comment:**

The paper "Geographical Erasure in Language Generation" presents work on the imbalance of generated data from LLMs, more concrete about the lack of representation of certain languages in generated data. The paper also presents a method to counter this phenomenon.

Arguments for rejecting the paper mentioned by the reviewers are related to the novelty of the task and details on basic assumptions.
Arguments for accepting the paper mentioned by the reviewers are related to the aspect studied (geographic erasure) and its potential to extend this to other similar phenomena.